# FAAH Inhibition Restores Early Life Stress-Induced Alterations in PFC microRNAs Associated with Depressive-Like Behavior in Male and Female Rats

**DOI:** 10.3390/ijms232416101

**Published:** 2022-12-17

**Authors:** Anna Portugalov, Hiba Zaidan, Inna Gaisler-Salomon, Cecilia J. Hillard, Irit Akirav

**Affiliations:** 1Department of Psychology, School of Psychological Sciences, University of Haifa, Haifa 3498838, Israel; 2The Integrated Brain and Behavior Research Center (IBBRC), University of Haifa, Haifa 3498838, Israel; 3Neuroscience Research Center, Department of Pharmacology and Toxicology, Medical College of Wisconsin, Milwaukee, WI 53226, USA

**Keywords:** cannabinoids, early life stress, depression, microRNAs, URB597

## Abstract

Early life stress (ELS) increases predisposition to depression. We compared the effects of treatment with the fatty acid amide hydrolase (FAAH) inhibitor URB597, and the selective serotonin reuptake inhibitor paroxetine, on ELS-induced depressive-like behavior and the expression of microRNAs (miRs) associated with depression in the medial prefrontal cortex (mPFC), hippocampal CA1 area, lateral habenula and dorsal raphe in rats. We also examined the mRNA expression of serotonergic (*htr1a* and *slc6a4*) and endocannabinoid (*cnr1*, *cnr2* and *faah*) targets in the mPFC following ELS and pharmacological treatment. Adult males and females exposed to the ‘Limited Bedding and Nesting’ ELS paradigm demonstrated a depressive-like phenotype and late-adolescence URB597 treatment, but not paroxetine, reversed this phenotype. In the mPFC, ELS downregulated miR-16 in males and miR-135a in females and URB597 treatment restored this effect. In ELS females, the increase in *cnr2* and decrease in *faah* mRNAs in the mPFC were reversed by URB597 treatment. We show for the first time that URB597 reversed ELS-induced mPFC downregulation in specific miRs and stress-related behaviors, suggesting a novel mechanism for the beneficial effects of FAAH inhibition. The differential effects of ELS and URB597 on males and females highlight the importance of developing sex-specific treatment approaches.

## 1. Introduction

Early life adversity is a major risk factor for various psychopathologies including major depressive disorder (MDD) [1]. Exposure to early life stress (ELS) has been associated with poor response to current antidepressant pharmacotherapies in adulthood [2], emphasizing the need for novel treatment approaches.

The endocannabinoid (ECB) system has been implicated in the etiology of depression, and ECB signaling enhancement may play a crucial role in the alleviation of depressive symptoms [3,4] and decreasing depressive-like behavior in animal models [5,6]. The ECB system is a neuromodulatory lipid system, which encompasses the cannabinoid receptors: CB1 (CB1r) and CB2 (CB2r); endogenous ligands: *N*-arachidonylethanolamine (anandamide; AEA) and 2-arachidonoylglycerol (2AG); and enzymes that degrade them: 2 monoacylglycerol lipase (MAGL) for 2-AG and fatty acid amide hydrolase (FAAH) for AEA. CB1r, AEA, and FAAH are present in brain areas involved in stress, emotions, and, MDD pathophysiology including the medial prefrontal cortex (mPFC), hippocampal CA1 area, and lateral habenula (LHb) [7,8,9]. These brain areas are also vulnerable to the detrimental effects of ELS [8,10,11].

We have recently found that ELS induced protracted depressive-like behavior and impaired cognition, and that these effects were restored by enhancing ECB signaling with the CB1/2 receptor agonist WIN55,212-2, the FAAH inhibitor URB597 (URB) or the MAGL inhibitor JZL-184 [12,13,14]. To induce ELS, we used the “Limited Bedding and Nesting (LBN)” paradigm, which causes fragmentation of maternal care by providing the dam with insufficient conditions for proper pup care [15].

Exposure to ELS can affect gene expression via epigenetic processes that regulate transcription [16]. In particular, microRNAs (miRNAs; miRs), a class of small non-coding RNAs that regulate gene expression by degrading their target messenger RNAs and/or inhibiting their translation, have been implicated in susceptibility to stress and depression [17,18,19]. Specifically, there is evidence that several miRs are affected by ELS. For example, hippocampal miR-16 was upregulated in adult males following maternal separation [20] and inescapable shock during early adolescence [21]; inescapable shock also downregulated PFC miR-135a expression. The above-mentioned studies were performed exclusively in male rodents, although stress affects men and women differently and women are diagnosed with stress-related disorders and depression more often than men [22,23,24]. Thus, research in both males and females is essential. In humans, mir-135a and miR-16 are implicated in ELS and depression via modulation of serotonergic transmission, affecting the expression of 5-HT auto-receptors 1a (5HT1A) and serotonin transporter (SERT) [25,26,27].

Here, we examined the potential antidepressant effects of URB, compared with the selective serotonin reuptake inhibitor (SSRI) paroxetine (PAR), on ELS-induced depressive-like phenotype in adult rats. We examined whether the antidepressant effects of URB are associated with alterations in miRNAs (miR-135a-5p and miR-16-5p) in the mPFC, CA1, LHb and dorsal raphe (DR). Additionally, we examined ELS-induced alterations in serotonergic receptor and transporter (5HT1A and SERT, respectively) and ECB (CB1r, CB2r and FAAH) targets in the mPFC. To the best of our knowledge, this study is the first to examine the impact of FAAH inhibition on the expression of miRNAs in brain areas associated with depression following exposure to ELS.

## 2. Results

### 2.1. The Effects of ELS and Chronic Late-Adolescence Treatment with URB597 or Paroxetine on the Behavior of Adult Males and Females

We examined the long-term effects of URB or PAR administered to ELS males and females during late adolescence on behavior in adulthood (see Figure 1a for experimental design, n = 7 of each sex in each group). We used a 2 × 2 × 3 design with main factors of Sex, ELS (NoELS/ELS) and Drug (Vehicle/URB/PAR). In cases of a significant sex effect or three-way interaction, data from male and female rats were analyzed separately.

For distance in the open field test (OFT) (Figure 1b), univariate ANOVA revealed significant effects of sex (F_(1,72)_ = 56.28, *p <* 0.001), ELS (F_(1,72)_ = 81.15, *p <* 0.001), drug (F_(2,72)_ = 17.44, *p <* 0.001), sex × drug (F_(2,72)_ = 5.41, *p =* 0.006) and ELS × drug × sex (F_(2,72)_ = 6.13, *p =* 0.003) interactions, with no effect of ELS × sex (F_(1,72)_ = 1.34, *p =* 0.24) and ELS × drug (F_(2,72)_ = 0.40, *p =* 0.66) interactions.

In males (Figure 1b, left), two-way ANOVA revealed significant effects of ELS (F_(1,36)_ = 26.80, *p <* 0.001) and drug (F_(2,36)_ = 12.48, *p <* 0.001), with no effect of ELS × drug (F_(2,36)_ = 2.37, *p =* 0.108) interaction. ELS males travelled less distance than NoELS, suggesting ELS-induced hypo-activity. Post hoc comparisons for drug revealed that URB males travelled more distance compared to males treated with vehicle (*p =* 0.001) or paroxetine (*p <* 0.001), suggesting that URB increased motor activity in NoELS and ELS groups.

In females (Figure 1b, right), two-way ANOVA revealed significant effects of ELS (F_(1,36)_ = 60.76, *p <* 0.001), drug (F_(2,36)_ = 9.99, *p <* 0.001) and ELS × drug (F_(2,36)_ = 4.48, *p =* 0.018) interaction. Post hoc comparisons revealed that NoELS-PAR females travelled less distance compared to NoELS-Vehicle (*p <* 0.001) and NoELS-URB (*p =* 0.007) groups, suggesting that PAR had an effect by itself on activity in non-stressed females.

For freezing in the OFT (Figure 1c), univariate ANOVA revealed significant effects of sex (F_(1,72)_ = 8.13, *p =* 0.006), ELS (F_(1,72)_ = 74.86, *p <* 0.001), drug (F_(2,72)_ = 5.23, *p =* 0.008) and ELS × drug (F_(2,72)_ = 3.60, *p =* 0.03) interaction, with no effect of ELS × sex (F_(1,72)_ = 2.95, *p =* 0.090), sex × drug (F_(2,72)_ = 0.73, *p =* 0.484) and ELS × drug × sex (F_(2,72)_ = 0.08, *p =* 0.920) interactions.

In males (Figure 1c, left), two-way ANOVA revealed significant effects of ELS (F_(1,36)_ = 45.46, *p <* 0.001) and drug (F_(2,36)_ = 4.11, *p =* 0.025), with no effect of ELS × drug (F_(2,36)_ = 1.67, *p =* 0.202) interaction. ELS males spent more time freezing compared to NoELS, suggesting anxiety-like behavior. URB males spent less time freezing compared to PAR males, suggesting decreased anxiety-like behavior (*p =* 0.018).

In females (Figure 1c, right), two-way ANOVA revealed significant effects of ELS (F_(1,36)_ = 29.41, *p <* 0.001), with no effect of drug (F_(2,36)_ = 1.34, *p =* 0.274) and ELS × drug (F_(2,36)_ = 2.08, *p =* 0.139) interaction. ELS females spent more time freezing compared to NoELS, suggesting increased anxiety-like behavior.

For time spent in the center in the OFT (Figure 1d), univariate ANOVA revealed significant effects of sex (F_(1,72)_ = 10.38, *p =* 0.002) and drug (F_(2,72)_ = 4.36, *p =* 0.016), with no effect of ELS (F_(1,72)_ = 0.22, *p =* 0.637), ELS × sex (F_(1,72)_ = 0.02, *p =* 0.886), sex × drug (F_(2,72)_ = 1.58, *p =* 0.213), ELS × drug (F_(2,72)_ = 0.02, *p =* 0.972) and ELS × drug × sex (F_(2,72)_ = 1.49, *p =* 0.232) interactions.

In males (Figure 1d, left), two-way ANOVA revealed significant effects of drug (F_(2,36)_ = 4.50, *p =* 0.018) and ELS × drug (F_(2,36)_ = 0.45, *p =* 0.636) interaction, with no effect of ELS (F_(1,36)_ = 0.15, *p =* 0.695). Post hoc comparisons revealed that PAR males spent less time in the center of the arena compared to males treated with vehicle (*p =* 0.030) or URB (*p =* 0.043), suggesting increased anxiety-like behavior in the PAR groups.

In females (Figure 1d, right), two-way ANOVA did not reveal any significant effects of ELS (F_(1,36)_ = 0.07, *p =* 0.792), drug (F_(2,36)_ = 0.54, *p =* 0.586) and ELS × drug (F_(2,36)_ = 1.23, *p =* 0.302) interaction.

For social preference (SP) (Figure 1e), univariate ANOVA revealed significant effects of sex (F_(1,72)_ = 9.63, *p =* 0.003), drug (F_(2,72)_ = 7.59, *p =* 0.001), ELS × sex (F_(1,72)_ = 30.09, *p <* 0.001), sex × drug (F_(2,72)_ = 6.41, *p =* 0.003) and ELS × drug (F_(2,72)_ = 14.17, *p <* 0.001) interactions, with no effect of ELS (F_(1,72)_ = 0.83, *p =* 0.364) and ELS × drug × sex (F_(2,72)_ = 1.87, *p =* 0.161) interaction.

In males (Figure 1e, left), two-way ANOVA revealed significant effects of ELS (F_(1,36)_ = 9.86, *p =* 0.003), drug (F_(2,36)_ = 10.14, *p <* 0.001) and ELS × drug (F_(2,36)_ = 4.75, *p =* 0.015) interaction. Post hoc comparisons revealed that ELS-URB males demonstrated increased SP compared to ELS-Vehicle (*p =* 0.002) and ELS-PAR (*p <* 0.001) groups, however a significant difference from the chance level (0.5 DI) was observed in all groups, suggesting intact SP in males. In NoELS males, the NoELS-PAR group demonstrated decreased SP compared to the NoELS-Vehicle (*p =* 0.008) group, suggesting that PAR had an effect by itself on SP in non-stressed males.

In females (Figure 1e, right), two-way ANOVA revealed significant effects of ELS (F_(1,36)_ = 21.78, *p <* 0.001), drug (F_(2,36)_ = 3.46, *p =* 0.042) and ELS × drug (F_(2,36)_ = 11.71, *p <* 0.001) interaction. Post hoc comparisons revealed that ELS-URB females demonstrated increased SP compared to ELS-Vehicle (*p =* 0.002), suggesting that URB restored the ELS-induced decrease in SP. Additionally, NoELS-URB females demonstrated decreased SP compared to NoELS-Vehicle (*p =* 0.025), suggesting that URB affected SP in non-stressed females.

For social recognition (SR) (Figure 1f), univariate ANOVA revealed significant effects of sex × drug (F_(2,72)_ = 4.07, *p =* 0.021) and ELS × drug (F_(2,72)_ = 22.78, *p <* 0.001) interactions, with no effect of sex (F_(1,72)_ = 0.01, *p =* 0.918), ELS (F_(1,72)_ = 0.005, *p =* 0.944), drug (F_(2,72)_ = 2.25, *p =* 0.112), ELS × sex (F_(1,72)_ = 3.52, *p =* 0.065) and ELS × drug × sex (F_(2,72)_ = 1.80, *p =* 0.172) interactions.

In males (Figure 1f, left), two-way ANOVA revealed significant effects of drug (F_(2,36)_ = 4.77, *p =* 0.014) and ELS × drug (F_(2,36)_ = 7.53, *p =* 0.002) interaction, with no effect of ELS (F_(1,36)_ = 1.62, *p =* 0.211). Post hoc comparisons revealed that ELS-URB males demonstrated an increased SR discrimination index compared to ELS-Vehicle (*p =* 0.035) and ELS-PAR (*p =* 0.003) groups, suggesting that URB restored the ELS-induced decrease in SR. In the NoELS groups, the NoELS-URB (*p =* 0.046) and NoELS-PAR (*p =* 0.048) groups demonstrated decreased SR compared to the NoELS-Vehicle group, suggesting that both URB and PAR had an effect by themselves on SR in non-stressed males.

In females (Figure 1f, right), two-way ANOVA revealed significant effects of ELS × drug (F_(2,36)_ = 18.97, *p <* 0.001) interaction, with no effect of ELS (F_(1,36)_ = 1.95, *p =* 0.170) and drug (F_(2,36)_ = 0.90, *p =* 0.414). Post hoc comparisons revealed that ELS-URB females demonstrated an increased SR compared to the ELS-Vehicle and ELS-PAR (*p <* 0.001) groups. This suggests that URB restored the ELS-induced decrease in SR. In the NoELS groups, the NoELS-URB group demonstrated decreased SR compared to the NoELS-PAR (*p =* 0.006) group.

To determine whether each group discriminates between the novel and the familiar juveniles, we performed a one-sample *t*-test on each group; in males, a significant difference from the 0.5 DI was observed in the NoELS-Vehicle (t_(6)_ = 2.53, *p =* 0.044) and ELS-URB (t_(6)_ = 4.57, *p =* 0.004), groups, but not in the other groups [NoELS-URB (t_(6)_ = −1.29, *p =* 0.244), NoELS-PAR (t_(6)_ = −0.87, *p =* 0.415), ELS-Vehicle (t_(6)_ = 0.46, *p =* 0.659),ELS-PAR (t_(6)_ = −1.40, *p =* 0.210)]. This suggests that only the NoELS-Vehicle and ELS-URB groups showed intact social short-term memory, hence, preferred the unfamiliar juvenile over the familiar.

In females, one-sample *t*-test performed on each group revealed a significant difference from the 0.5 DI only in the ELS-URB (t_(6)_ = 6.34, *p =* 0.001) and NoELS-PAR groups (t_(6)_ = 3.92, *p =* 0.008), suggesting that only these groups showed intact performance [NoELS-Vehicle (t_(6)_ = 1.58, *p =* 0.163), NoELS-URB (t_(6)_ = −1.31, *p =* 0.238), ELS-Vehicle (t_(6)_ = −1.52, *p =* 0.178) and ELS-PAR (t_(6)_ = −2.12, *p =* 0.078)].

We also measured the total time of exploration in the social tests (see Appendix A).

For the forced swim test (FST) (Figure 1f), univariate ANOVA revealed significant effects of sex (F_(1,72)_ = 12.57, *p =* 0.001), ELS (F_(1,72)_ = 5.64, *p =* 0.020), drug (F_(2,72)_ = 13.21, *p <* 0.001) and ELS × drug (F F_(2,72)_ = 13.94, *p <* 0.001) interactions, with no effect of ELS × sex (F_(1,72)_ = 2.08, *p =* 0.153), sex × drug (F_(2,72)_ = 0.17, *p =* 0.840) and ELS × drug × sex (F_(2,72)_ = 0.34, *p =* 0.712) interactions.

In males (Figure 1g, left), two-way ANOVA revealed significant effects of ELS (F_(1,36)_ = 12.52, *p =* 0.001), drug (F_(2,36)_ = 9.03, *p =* 0.001) and ELS × drug (F_(2,36)_ = 11.47, *p <* 0.001) interaction. Post hoc comparisons revealed that ELS-URB males demonstrated decreased immobility compared to the ELS-Vehicle and ELS-PAR (*p =* 0.040) groups, suggesting that URB restored the ELS-induced increase in immobility. In the NoELS groups, the NoELS-URB (*p =* 0.006) and NoELS-PAR (*p <* 0.001) groups demonstrated increased immobility compared to the NoELS-Vehicle group, suggesting that both URB and PAR had an effect by themselves on learned helplessness in non-stressed males.

In females (Figure 1g, right), two-way ANOVA revealed significant effects of drug (F_(2,36)_ = 5.73, *p =* 0.007) and ELS × drug (F_(2,36)_ = 5.36, *p =* 0.009) interaction, with no effect of ELS (F_(1,36)_ = 0.30, *p =* 0.584). Post hoc comparisons revealed that ELS-URB females demonstrated decreased immobility compared to ELS-Vehicle (*p =* 0.028) and ELS-PAR (*p =* 0.004) groups suggesting that URB restored the ELS-induced increase in immobility. In the NoELS groups, NoELS-PAR demonstrated increased immobility compared to NoELS-Vehicle (*p =* 0.021), suggesting that PAR had an effect by itself on learned helplessness in non-stressed females.

To summarize, URB restored the ELS-induced decrease in SP in females and restored the ELS-induced decrease in SR and increase in immobility in the FST in both males and females.

### 2.2. The Effects of ELS and Chronic Late-Adolescence Treatment with URB or PAR on the Expression of miR-135a-5p in Adult Males and Females

In the mPFC (Figure 2a), univariate ANOVA revealed significant effects of sex (F_(1,65)_ = 2341.27, *p <* 0.001), ELS (F_(1,65)_ = 409.86, *p <* 0.001), drug (F_(2,65)_ = 35.89, *p <* 0.001), sex × drug (F_(2,65)_ = 29.20, *p <* 0.001), ELS × drug (F_(2,65)_ = 72.28, *p <* 0.001) and ELS × drug × sex (F_(2,65)_ = 24.58, *p <* 0.001) interactions, with no effect of ELS × sex (F_(1,65)_ = 2.03, *p =* 0.159) interaction.

In males (Figure 2a, left), two-way ANOVA revealed significant effects of ELS (F_(1,27)_ = 280.57, *p <* 0.001), drug (F_(2,27)_ = 34.45, *p <* 0.001) and ELS × drug (F_(2,27)_ = 102.52, *p <* 0.001) interaction. Post hoc comparisons revealed decreased miR-135a in ELS-PAR compared to the ELS-Vehicle and ELS-URB (*p <* 0.001) groups. In the NoELS groups, miR-135a expression was upregulated in the NoELS-URB and NoELS-PAR groups compared to the NoELS-Vehicle (*p <* 0.001) group.

In females (Figure 2a, right), two-way ANOVA revealed significant effects of ELS (F_(1,38)_ = 170.23, *p <* 0.001), drug (F_(2,38)_ = 35.38, *p <* 0.001) and ELS × drug (F_(2,38)_ = 10.40, *p <* 0.001) interaction. Post hoc comparisons revealed upregulation in miR-135a in ELS-URB compared to the ELS-Vehicle (*p =* 0.009), suggesting that URB normalized ELS-induced decrease in the expression of miR-135a. In the NoELS groups, miR-135a was upregulated in the NoELS-URB and NoELS-PAR groups compared to the NoELS-Vehicle (*p <* 0.001) group.

In the CA1 (Figure 2b), univariate ANOVA revealed significant effects of sex (F_(1,66)_ = 255.62, *p <* 0.001), ELS (F_(1,66)_ = 8.95, *p =* 0.004), drug (F_(2,66)_ = 10.01, *p <* 0.001), ELS × sex (F_(1,66)_ = 13.33, *p =* 0.001), sex × drug (F_(2,66)_ = 18.31, *p <* 0.001), ELS × drug (F_(2,66)_ = 3.53, *p =* 0.035) and ELS × drug × sex (F_(2,66)_ = 10.31, *p <* 0.001) interactions.

In males (Figure 2b, left), two-way ANOVA revealed significant effects of drug (F_(2,31)_ = 22.87, *p <* 0.001) and ELS × drug (F_(2,31)_ = 5.71, *p =* 0.008) interaction, with no effect of ELS (F_(1,31)_ = 0.19, *p =* 0.664). Post hoc comparisons revealed decreased miR-135a in ELS-PAR compared to ELS-Vehicle (*p <* 0.001) and ELS-URB (*p =* 0.001) groups. In the NoELS groups, miR-135a was downregulated in the NoELS-URB (*p =* 0.003) and NoELS-PAR (*p <* 0.001) groups compared to the NoELS-Vehicle group.

In females (Figure 2b, right), two-way ANOVA revealed significant effects of ELS (F_(1,35)_ = 24.89, *p <* 0.001), drug (F_(2,35)_ = 3.59, *p =* 0.038) and ELS × drug (F_(2,35)_ = 8.27, *p =* 0.001) interaction. Post hoc comparisons revealed an upregulation of miR-135a in NoELS-URB and NoELS-PAR compared to NoELS-Vehicle (*p =* 0.003), suggesting that URB and PAR affected CA1 miR-135a expression in non-stressed females.

In the LHb (Figure 2c), univariate ANOVA revealed significant effects of ELS (F_(1,63)_ = 45.29, *p <* 0.001), drug (F_(2,63)_ = 64.37, *p <* 0.001), ELS × sex (F_(1,63)_ = 156.14, *p <* 0.001), sex × drug (F_(2,63)_ = 7.07, *p =* 0.002), ELS × drug (F_(2,63)_ = 24.68, *p <* 0.001) and ELS × drug × sex (F_(2,63)_ = 16.95, *p <* 0.001) interactions, with no effect of sex (F_(1,63)_ = 0.72, *p =* 0.399).

In males (Figure 2c, left), two-way ANOVA revealed significant effects of ELS (F_(1,28)_ = 10.90, *p =* 0.003), drug (F_(2,28)_ = 10.89, *p <* 0.001) and ELS × drug (F_(2,28)_ = 5.05, *p =* 0.013) interaction. Post hoc comparisons revealed decreased expression of miR-135a in ELS-PAR compared to ELS-Vehicle (*p =* 0.009) and ELS-URB (*p =* 0.014) groups. In the NoELS groups, miR-135a was downregulated in the NoELS-URB (*p =* 0.001) and NoELS-PAR (*p <* 0.001) groups compared to the NoELS-Vehicle group.

In females (Figure 2c, right), two-way ANOVA revealed significant effects of ELS (F_(1,35)_ = 300.58, *p <* 0.001), drug (F_(2,35)_ = 89.19, *p <* 0.001) and ELS × drug (F_(2,35)_ = 53.88, *p <* 0.001) interaction. Post hoc comparisons revealed decreased miR-135a in ELS-PAR compared to ELS-Vehicle (*p =* 0.008) and ELS-URB (*p =* 0.001) groups. In the NoELS groups, miR-135a was downregulated in the NoELS-URB and NoELS-PAR groups compared to the NoELS-Vehicle (*p <* 0.001) group.

In the DR (Figure 2d), univariate ANOVA revealed significant effects of sex (F_(1,56)_ = 12,047.31, *p <* 0.001), ELS (F_(1,56)_ = 101.85, *p <* 0.001), drug (F_(2,56)_ = 204.25, *p <* 0.001), ELS × sex (F_(1,56)_ = 232.49, *p <* 0.001), sex × drug (F_(2,56)_ = 17.45, *p <* 0.001), ELS × drug (F_(2,56)_ = 53.17, *p <* 0.001) and ELS × drug × sex (F_(2,56)_ = 49.05, *p <* 0.001) interactions.

In males (Figure 2d, left), two-way ANOVA revealed significant effects of ELS (F_(1,30)_ = 10.47, *p =* 0.003), drug (F_(2,30)_ = 89.07, *p <* 0.001) and ELS × drug (F_(2,30)_ = 43.41, *p <* 0.001) interaction. Post hoc comparisons revealed decreased miR-135a in ELS-PAR compared to ELS-Vehicle and ELS-URB (*p <* 0.001) groups. Additionally, miR-135a was downregulated in the NoELS-URB and NoELS-PAR groups compared to the NoELS-Vehicle (*p <* 0.001) group.

In females (Figure 2d, right), two-way ANOVA revealed significant effects of ELS (F_(1,26)_ = 498.59, *p <* 0.001), drug (F_(2,26)_ = 160.88, *p <* 0.001) and ELS × drug (F_(2,26)_ = 63.64, *p <* 0.001) interaction. Post hoc comparisons revealed decreased miR-135a in ELS-URB (*p =* 0.006) and ELS-PAR (*p <* 0.001) compared to the ELS-Vehicle group. In the NoELS groups, miR-135a was downregulated in the NoELS-URB and NoELS-PAR groups compared to the NoELS-Vehicle (*p <* 0.001) group.

To summarize, ELS decreased miR-135a expression in the mPFC and increased its expression in the DR of female rats. In both regions, this effect was reversed by the late-adolescence URB administration. In the DR, it was also downregulated by PAR treatment. URB and PAR were also found to affect non-stressed males and females. URB and PAR upregulated miR-135a expression in the mPFC of NoELS males and females and the CA1 of NoELS females. Conversely, in the CA1 of NoELS males and the LHb and DR of NoELS males and females, URB and PAR downregulated the expression of miR-135a.

### 2.3. The Effects of ELS and Chronic Late-Adolescence Treatment with URB or PAR on the Expression of miR-16-5p in Adult Males and Females

In the mPFC (Figure 3a), univariate ANOVA revealed significant effects of sex (F_(1,69)_ = 284.28, *p <* 0.001), ELS (F_(1,69)_ = 60.65, *p <* 0.001), drug (F_(2,69)_ = 17.65, *p <* 0.001), ELS × sex (F_(1,69)_ = 52.81, *p <* 0.001), sex × drug (F_(2,69)_ = 10.63, *p <* 0.001), ELS × drug (F_(2,69)_ = 8.71, *p <* 0.001) and ELS × drug × sex (F_(2,69)_ = 4.06, *p =* 0.021) interactions.

In males (Figure 3a, left), two-way ANOVA revealed significant effects of ELS (F_(1,30)_ = 113.04, *p <* 0.001), drug (F_(2,30)_ = 28.80, *p <* 0.001) and ELS × drug (F_(2,30)_ = 12.25, *p <* 0.001) interaction. Post hoc comparisons revealed increased miR-16 in ELS-URB compared to ELS-Vehicle (*p =* 0.026), suggesting that URB normalized the ELS-induced decrease in miR-16. In the NoELS group, miR-16 was upregulated in the NoELS-URB and NoELS-PAR (*p <* 0.001) groups compared to NoELS-Vehicle.

In females (Figure 3a, right), two-way ANOVA did not reveal any significant effects of ELS (F_(1,39)_ = 0.14, *p =* 0.710), drug (F_(2,39)_ = 0.78, *p =* 0.462) and ELS × drug (F_(2,39)_ = 0.64, *p =* 0.533) interaction.

In the CA1 (Figure 3b), univariate ANOVA revealed significant effects of sex (F_(1,62)_ = 67.60, *p <* 0.001), ELS (F_(1,62)_ = 92.75, *p <* 0.001), drug (F_(2,62)_ = 10.40, *p <* 0.001), sex × drug (F_(2,62)_ = 11.04, *p <* 0.001), ELS × drug (F_(2,62)_ = 30.23, *p <* 0.001) and ELS × drug × sex (F_(2,62)_ = 10.72, *p <* 0.001) interactions, with no effect of ELS × sex (F_(1,62)_ = 1.40, *p =* 0.241) interaction.

In males (Figure 3b, left), two-way ANOVA revealed significant effects of ELS (F_(1,31)_ = 26.19, *p <* 0.001), drug (F_(2,31)_ = 5.50, *p =* 0.009) and ELS × drug (F_(2,31)_ = 25.83, *p <* 0.001) interaction. Post hoc comparisons revealed decreased miR-16 in ELS-PAR compared to the ELS-Vehicle and ELS-URB (*p <* 0.001) groups. In the NoELS groups, miR-16 was upregulated in the NoELS-PAR group compared to the NoELS-Vehicle (*p =* 0.012) group.

In females (Figure 3b, right), two-way ANOVA revealed significant effects of ELS (F_(1,31)_ = 92.49, *p <* 0.001), drug (F_(2,31)_ = 21.82, *p <* 0.001) and ELS × drug (F_(2,31)_ = 11.78, *p <* 0.001) interaction. Post hoc comparisons revealed an upregulation of miR-16 in NoELS-URB and NoELS-PAR compared to NoELS-Vehicle (*p <* 0.001).

In the LHb (Figure 3c), univariate ANOVA revealed significant effects of sex (F_(1,59)_ = 110.96, *p <* 0.001), ELS (F_(1,59)_ = 157.13, *p <* 0.001), drug (F_(2,59)_ = 28.30, *p <* 0.001), ELS × sex (F_(1,59)_ = 95.46, *p <* 0.001), sex × drug (F_(2,59)_ = 16.87, *p <* 0.001), ELS × drug (F_(2,59)_ = 64.58, *p <* 0.001) and ELS × drug × sex (F_(2,59)_ = 18.69, *p <* 0.001) interactions.

In males (Figure 3c, left), two-way ANOVA revealed significant effects of ELS (F_(1,26)_ = 4.32, *p =* 0.048) and ELS × drug (F_(2,26)_ = 17.36, *p <* 0.001) interaction, with no effect of drug (F_(2,26)_ = 1.53, *p =* 0.234). Post hoc comparisons revealed increased miR-16 in ELS-URB (*p =* 0.041) and ELS-PAR (*p <* 0.001) compared to ELS-Vehicle. In the NoELS groups, miR-16 was downregulated in the NoELS-PAR group compared to the NoELS-Vehicle (*p =* 0.004) group.

In females (Figure 3c, right), two-way ANOVA revealed significant effects of ELS (F_(1,33)_ = 238.01, *p <* 0.001), drug (F_(2,33)_ = 41.70, *p <* 0.001) and ELS × drug (F_(2,33)_ = 64.98, *p <* 0.001) interaction. Post hoc comparison revealed decreased miR-16 in NoELS-URB and NoELS-PAR compared to NoELS-Vehicle (*p <* 0.001).

In the DR (Figure 3d), univariate ANOVA revealed significant effects of sex (F_(1,64)_ = 3167.02, *p <* 0.001), ELS (F_(1,64)_ = 26.52, *p <* 0.001), drug (F_(2,64)_ = 25.28, *p <* 0.001), sex × drug (F_(2,64)_ = 21.97, *p <* 0.001), ELS × drug (F_(2,64)_ = 32.41, *p <* 0.001) and ELS × drug × sex (F_(2,64)_ = 9.35, *p <* 0.001) interactions, with no effect of ELS × sex (F_(1,64)_ = 2.14, *p =* 0.148) interaction.

In males (Figure 3d, left), two-way ANOVA revealed significant effects of ELS (F_(1,35)_ = 5.91, *p =* 0.020), drug (F_(2,35)_ = 39.50, *p <* 0.001) and ELS × drug (F_(2,35)_ = 27.52, *p <* 0.001) interaction. Post hoc comparison revealed increased miR-16 in ELS-PAR compared to the ELS-Vehicle and ELS-URB (*p <* 0.001) groups.

In females (Figure 3d, right), two-way ANOVA revealed significant effects of ELS (F_(1,29)_ = 29.008, *p <* 0.001) and ELS × drug (F_(2,29)_ = 10.35, *p <* 0.001) interaction, with no effect of drug (F_(2,29)_ = 0.48, *p =* 0.623). Post hoc comparison revealed increased miR-16 in ELS-URB (*p =* 0.011) and ELS-PAR (*p =* 0.002) compared to ELS-Vehicle.

To summarize, URB restored the ELS-induced decrease in miR-16 expression in the mPFC and LHb of male rats. URB and PAR restored ELS-induced decrease in miR-16 in the LHb of males. URB and PAR also affected miR-16 in non-stressed rats. Both drugs upregulated miR-16 expression in the mPFC of NoELS males and the CA1 of NoELS females. Different drug effects were observed in NoELS and ELS rats in the CA1 and LHb; also, in these regions PAR and URB had opposite effects on miR-16 expression.

### 2.4. The Effects of ELS and Chronic Late-Adolescence Treatment with URB or PAR on mRNA Expression of Serotonergic and Endocannabinoid Genes in the mPFC in Adult Males and Females

In the mPFC, URB597 prevented the ELS-induced decrease in the expression of miR-135a and miR-16 in females and males, respectively. Hence, we further examined possible serotonergic (*htr1a* and *slc6a4*) and endocannabinoid (*cnr1*, *cnr2*, *faah*) mRNA targets in the mPFC of adult males and females (see Figure 1a for experimental design).

For *htr1a* (Figure 4a), univariate ANOVA revealed significant effects of sex (F_(1,55)_ = 104.48, *p <* 0.001), ELS (F_(1,55)_ = 19.10, *p <* 0.001), drug (F_(2,55)_ = 10.57, *p <* 0.001), sex × drug (F_(2,55)_ = 15.85, *p <* 0.001), ELS × drug (F_(2,55)_ = 60.46, *p <* 0.001) and ELS × drug × sex (F_(2,55)_ = 59.10, *p <* 0.001) interactions, with no effect of ELS × sex (F_(1,55)_ = 0.87, *p =* 0.354) interaction.

In males (Figure 4a, left), two-way ANOVA revealed significant effects of ELS (F_(1,28)_ = 5.50, *p =* 0.026), drug (F_(2,28)_ = 24.40, *p <* 0.001) and ELS × drug (F_(2,28)_ = 106.90, *p <* 0.001) interaction. Post hoc comparisons revealed decreased *htr1a* in ELS-Vehicle and ELS-URB compared to ELS-PAR (*p <* 0.001). In the NoELS groups, *htr1a* was downregulated in the NoELS-URB and NoELS-PAR groups compared to the NoELS-Vehicle (*p <* 0.001) group.

In females (Figure 4a, right), two-way ANOVA revealed significant effects of ELS (F_(1,27)_ = 15.28, *p <* 0.001) and ELS × drug (F_(2,27)_ = 6.71, *p =* 0.004) interaction, with no effect of drug (F_(2,27)_ = 0.57, *p =* 0.568). Post hoc comparisons revealed increased *htr1a* in NoELS-URB compared to NoELS-Vehicle (*p =* 0.037).

For *slc6a4* (Figure 4b)*,* univariate ANOVA revealed significant effects of sex (F_(1,49)_ = 3048.55, *p <* 0.001), drug (F_(2,49)_ = 20.58, *p <* 0.001), ELS × sex (F_(1,49)_ = 8.03, *p =* 0.007), sex × drug (F_(2,49)_ = 7.06, *p =* 0.002), ELS × drug (F_(2,49)_ = 10.99, *p <* 0.001) and ELS × drug × sex (F_(2,49)_ = 7.65, *p =* 0.001) interactions, with no effect of ELS (F_(1,49)_ = 1.81, *p =* 0.185).

In males (Figure 4b, left), two-way ANOVA revealed significant effects of drug (F_(2,24)_ = 15.44, *p <* 0.001) and ELS × drug (F_(2,24)_ = 11.46, *p <* 0.001) interaction, with no effect of ELS (F_(1,24)_ = 0.75, *p =* 0.395). Post hoc comparisons revealed increased expression of *slc6a4* in ELS-PAR compared to ELS-Vehicle and ELS-URB (*p <* 0.001). In the NoELS groups, *slc6a4* expression was downregulated in the NoELS-URB group compared to the NoELS-Vehicle (*p =* 0.020) group.

In females (Figure 4b, right), two-way ANOVA revealed significant effects of ELS (F_(1,25)_ = 15.67, *p =* 0.001) and drug (F_(2,25)_ = 9.37, *p =* 0.001), with no effect of ELS × drug (F_(2,24)_ = 11.46, *p <* 0.001) interaction. *Slc6a4* upregulation was found in ELS- and NoELS-PAR groups compared to NoELS-Vehicle (*p <* 0.001) and NoELS-URB (*p =* 0.050) groups, respectively.

For *cnr1* (Figure 4c)*,* univariate ANOVA revealed significant effects of sex (F_(1,71)_ = 22.84, *p <* 0.001), ELS (F_(1,71)_ = 178.19, *p <* 0.001), drug (F_(2,71)_ = 69.15, *p <* 0.001), ELS × sex (F_(1,71)_ = 80.98, *p <* 0.001), sex × drug (F_(2,71)_ = 46.58, *p <* 0.001), ELS × drug (F_(2,71)_ = 51.71, *p <* 0.001) and ELS × drug × sex (F_(2,71)_ = 30.39, *p <* 0.001) interactions.

In males (Figure 4c, left), two-way ANOVA revealed significant effects of ELS (F_(1,32)_ = 345.02, *p <* 0.001), drug (F_(2,32)_ = 162.27, *p <* 0.001) and ELS × drug (F_(2,32)_ = 114.35, *p <* 0.001) interaction. Post hoc comparisons revealed decreased *cnr1* in the ELS-Vehicle and ELS-URB groups compared to the ELS-PAR (*p <* 0.001) group. In the NoELS groups, *cnr1* was downregulated in the NoELS-URB group compared to the NoELS-Vehicle (*p =* 0.007) and NoELS-PAR (*p =* 0.002) groups.

In females (Figure 4c, right), two-way ANOVA revealed significant effects of ELS (F_(1,39)_ = 8.11, *p =* 0.007), with no effect of drug (F_(2,39)_ = 2.66, *p =* 0.082) and ELS × drug (F_(2,39)_ = 1.26, *p =* 0.293) interaction. *cnr1* downregulation was found in the ELS groups compared to the NoELS groups.

For *cnr2* (Figure 4d)*,* univariate ANOVA revealed significant effects of sex (F_(1,61)_ = 353.05, *p <* 0.001), drug (F_(2,61)_ = 90.65, *p <* 0.001), ELS × sex (F_(1,61)_ = 111.54, *p <* 0.001), sex × drug (F_(2,61)_ = 46.58, *p <* 0.001), ELS × drug (F_(2,61)_ = 49.86, *p <* 0.001) and ELS × drug × sex (F_(2,61)_ = 201.64, *p <* 0.001) interactions, with no effect of ELS (F_(1,61)_ = 0.03, *p =* 0.849).

In males (Figure 4d, left), two-way ANOVA revealed significant effects of ELS (F_(1,30)_ = 55.09, *p <* 0.001), drug (F_(2,30)_ = 145.92, *p <* 0.001) and ELS × drug (F_(2,30)_ = 438.73, *p <* 0.001) interaction. Post hoc comparisons revealed decreased *cnr2* in ELS-Vehicle and ELS-URB males compared to ELS-PAR (*p <* 0.001). In the NoELS groups, *cnr2* was downregulated in the NoELS-Vehicle and NoELS-URB groups compared to the NoELS-PAR (*p <* 0.001) group.

In females (Figure 4d, right), two-way ANOVA revealed significant effects of ELS (F_(1,31)_ = 56.54, *p <* 0.001) and drug (F_(2,31)_ = 3.43, *p =* 0.045), with no effect of ELS × drug (F_(2,31)_ = 1.65, *p =* 0.208) interaction. *cnr2* upregulation was found in the ELS groups compared to the NoELS groups, and downregulation in the URB groups compared to the vehicle groups (*p <* 0.017).

For *faah* (Figure 4e)*,* univariate ANOVA revealed significant effects of sex (F_(1,58)_ = 236.21, *p <* 0.001), ELS (F_(1,58)_ = 108.32, *p <* 0.001), drug (F_(2,58)_ = 41.12, *p <* 0.001), ELS × sex (F_(1,58)_ = 12.88, *p =* 0.001), sex × drug (F_(2,58)_ = 32.71, *p <* 0.001), ELS × drug (F_(2,58)_ = 40.95, *p <* 0.001) and ELS × drug × sex (F_(2,58)_ = 22.63, *p <* 0.001) interactions.

In males (Figure 4e, left), two-way ANOVA revealed significant effects of ELS (F_(1,29)_ = 78.5, *p <* 0.001), drug (F_(2,29)_ = 52.71, *p <* 0.001) and ELS × drug (F_(2,29)_ = 49.8, *p <* 0.001) interaction. Post hoc comparisons revealed decreased expression of *faah* in ELS-Vehicle and ELS-URB males compared to ELS-PAR (*p <* 0.001).

In females (Figure 4e, right), two-way ANOVA revealed significant effects of ELS (F_(1,29)_ = 31, *p <* 0.001), drug (F_(2,29)_ = 12.88, *p <* 0.001) and ELS × drug (F_(2,29)_ = 3.82, *p =* 0.034) interaction. Post hoc comparisons revealed decreased expression of *faah* in ELS-Vehicle females compared to ELS-URB (*p <* 0.001) and ELS-PAR (*p =* 0.008) groups.

To summarize, ELS decreased *faah* expression and increased *cnr2* expression in the mPFC of female rats. These effects were reversed by the late-adolescence URB administration. ELS and URB decreased *htr1a, scl6e4, cnr1, cnr2 and faah* expression in males. Additionally, URB and PAR downregulated *htr1a* and *cnr2* in NoELS males.

### 2.5. Correlations between the Expression of miRNAs (miR-135a & miR-16) and Behavior in Adult Males and Females

Pearson bivariate correlations tests were conducted between the behavioral tests in adult males and females and the expression of miRNAs (miR-135a, Table 1; miR-16, Table 2) in all brain regions tested (mPFC, CA1, LHb and DR) to explore the association between the anxiogenic- and depressive-like phenotype of the rats and their miR expression.

For miR-135a, in the mPFC, significant positive correlations were found between miR-135a expression, and distance travelled (males: r = 0.62, *p =* 0.001; females: r = 0.35, *p =* 0.043) and a negative correlation with freezing (males: r = −0.57, *p =* 0.002; females: r = −0.37, *p =* 0.030) in the OFT, suggesting that decreased mPFC-miR-135a is associated with decreased activity and increased anxiety-like behavior in both sexes.

In the CA1, significant positive correlations were found only in males between miR-135a and SR (r = 0.51, *p =* 0.003) and time spent in the center in the OF (r = 0.40, *p =* 0.022), suggesting that increased CA1-miR-135a expression is associated with intact social short-term memory and decreased anxiety-like behavior in males.

In the LHb, significant positive correlations were found between miR-135a and SR in males (r = 0.41, *p =* 0.024) and time spent in the center in both males (r = 0.36, *p =* 0.049) and females (r = 0.36, *p =* 0.045). Negative correlations were found between miR-135a and freezing in males only (r = −0.64, *p <* 0.001). This suggests that increased LHb-miR-135a is associated with intact social short-term memory in males and with decreased anxiety-like behavior in both sexes.

In the DR, significant positive correlations were found between miR-135a expression and SR (r = 0.41, *p =* 0.025), travelled distance (r = 0.48, *p =* 0.006) and time spent in the center (r = 51, *p =* 0.004) in males; and between miR-135a and freezing in females (r = 0.45, *p =* 0.025). Negative correlations were observed between miR-135a and SP (r = −0.50, *p =* 0.016) in females. This suggests that increased DR- miR-135 is associated with intact social short-term memory, increased motor activity and decreased anxiety-like behavior in males, whereas in females, increased miR-135a is associated with increased anxiety-like behavior, decreased SP and hypo-activity.

For miR-16, in the mPFC, a significant positive correlation was found between miR-16 and distance travelled in the OF in males (r = 0.67, *p <* 0.001). Negative correlations were found between the miR and SP (r = −0.45, *p =* 0.017) and freezing in the OF in males (r = −0.57, *p =* 0.001). This suggests that decreased mPFC-miR-16 expression is associated with hypo-activity, increased SP, and increased anxiety-like behavior in males.

In the CA1, a significant positive correlation was found between miR-16 and distance travelled in the OF in males (r = 0.55, *p =* 0.001). Negative correlations were found between the miR and SP (r = −0.38, *p =* 0.043) in males, and with freezing in the OF in both males and females (males: r = −0.40, *p =* 0.018; females: r = −0.47, *p =* 0.015). This suggests that decreased CA1-miR-16 expression is associated with hypo-activity and increased social behavior in males, and increased anxiety-like behavior in both sexes.

In the LHb, a significant positive correlation was found between miR-16 and freezing in the OF (r = 0.46, *p =* 0.011) and a negative correlation with distance travelled in the OF (r = −0.42, *p =* 0.022) in females, suggesting that decreased LHb-miR-16 is associated with decreased anxiety-like behavior and increased activity in females.

In the DR, a significant positive correlation was found between miR-16 and freezing in females (r = 0.53, *p =* 0.005). Negative correlations were found between the miR and SR in males (r = −0.46, *p =* 0.007), and with distance travelled in the OF in both sexes (males: r = −0.40, *p =* 0.019; females: r = −0.55; *p =* 0.003). This suggests that decreased DR-miR-16 expression is associated with increased SR and increased activity in males, and with decreased anxiety-like behavior and increased activity in females.

### 2.6. Correlations between the Expression of Serotonergic and ECB mRNAs in the mPFC and Behavior in Adult Males and Females

Pearson bivariate correlations tests were conducted between the behavioral tests in adult males and females and the expression of mRNA serotonergic and ECB targets (*htr1a*, *slc6a4*, *cnr1*, *cnr2* and *faah*; Table 3) in the mPFC to explore the association between the anxiety- and depressive-like phenotype of the rats and mRNAs expression.

A significant positive correlation was found between *htr1a* mRNA and SR in males (r = 0.50, *p =* 0.011). Negative correlations were found between the gene and immobility ratio (r = −0.65, *p =* 0.001) and distance travelled in the OF in males (r = −0.62, *p =* 0.002); and between *htr1a* expression and SP in females (r = −0.46, *p =* 0.018). This suggests that decreased mPFC-*htr1a* is associated with impaired SR, increased immobility, and increased activity in males; and is associated with increased SP in females.

For *Scl6a4*, a significant correlation was found with freezing in the OF (r = 0.46, *p =* 0.040) and a negative correlation was found with SP (r = −0.46, *p =* 0.41) in females. This suggests that decreased mPFC-*Scl6a4* expression is associated with decreased anxiety-like behavior and increased SP in females.

For *cnr1,* significant negative correlations were found with SP (r = −0.47, *p =* 0.020) and SR (r = −0.56, *p =* 0.003) in males, suggesting that decreased mPFC-*cnr1* expression is associated with increased SP and SR in males.

For *cnr2,* a significant positive correlation was found with freezing (r = 0.44, *p =* 0.036), and a negative correlation was found with distance travelled in the OF in females (r = −0.53, *p =* 0.008), suggesting that decreased mPFC-*cnr2* is associated with increased activity and decreased anxiety-like behavior in females.

For *faah*, a significant positive correlation was found with SP in females (r = 0.46, *p =* 0.025). Negative significant correlations were found between the gene and SP (r = −0.49, *p =* 0.019) and SR (r = −0.52, *p =* 0.008) in males. This suggests that decreased mPFC-*faah* is associated with increased SP and SR in males, and with decreased SP in females.

### 2.7. Correlations between the Expression of miR-135a & miR-16 and Serotonergic and ECB mRNAs in the mPFC in Adult Males and Females

We also conducted Pearson bivariate correlations tests between the expression of miRNAs (miR-135a and miR-16) and their possible serotonergic (htr1a, slc6a4; [26,28,29]; and ECB targets (*cnr1*, *cnr2* and *faah*) in the mPFC in males and females (Table 4).

A significant positive correlation was found between miR-135a and *faah* expression in females (r = 0.54, *p =* 0.004), and negative correlations with *htr1a* (r = −0.55, *p =* 0.005) and *cnr2* (r = −0.51, *p =* 0.012) in males and *CNR2* in females (r = −0.76, *p <* 0.001). This suggests that increased miR-135a expression is associated with increased *faah* in females, decreased *htr1a* in males and decreased *cnr2* in both males and females.

For miR-16, a significant negative correlation was found with *scl6a4* in females (r = −0.49, *p* = 0.018). Additionally, a significant positive correlation was found between miR-135a and miR-16 in the mPFC in males (r = 0.71, *p <* 0.001). These correlations may suggest a possible association between the miRs.

## 3. Discussion

Exposure to ELS has detrimental protracted effects on behavior, the expression of miRNAs, and their targets in both sexes. Here, we show for the first time that FAAH inhibitor URB597 can restore ELS-induced decrease in mPFC miR-135a in females and miR-16 in males and the associated depressive-like phenotype in both sexes.

### 3.1. Effects on Behavior

URB597 administered during late adolescence restored the ELS-induced depressive-like phenotype in males and females; in males, URB597 restored ELS-induced hypo-activity and freezing in the OFT, decreased SR and learned helplessness in the FST. In females, URB597 restored an ELS-induced decrease in SR as well as learned helplessness. This corroborates with previous studies from our lab demonstrating that URB597 prevents the development of a depressive-like phenotype following ELS exposure [13,14].

Paroxetine did not prevent the ELS-induced depressive-like phenotype. The dose and duration used in our study were based on the literature [21]. In Liu et al. [21], paroxetine administered during P33–48 restored shock-induced anxiety-like behavior and spatial memory; however, the shocked rats were tested 24 h after the last injection [21] whereas in our study, rats were tested 30 days after the last injection. It is possible that our rats experienced withdrawal symptoms or that paroxetine needs to be present in the system during behavioral testing to be effective. Indeed, paroxetine was found as the most prevalent antidepressant with withdrawal symptoms [30].

### 3.2. Effects on miRNAs and Their Targets

ELS and the pharmacological treatment affected miR-135a and miR-16 expression in a sex- and region-dependent manner. These findings are in line with other studies demonstrating that these miRs are correlated with anxiety- and depressive-like phenotypes and are significantly downregulated or upregulated following stress exposure, depending upon the brain region studied and the type of stressor [17,20,22,24,31,32].

Interestingly, URB597 and paroxetine had pronounced effects on miRNAs in non-stressed males and females. However, in the stressed rats, they showed differential effects, probably due to the interaction with stress regulatory mechanisms. The most robust example is that in the non-stressed groups both URB597 and paroxetine upregulated mPFC-miR-16 in males and mPFC-miR-135a in females, but in the stressed groups, only URB597 restored their expression. This may suggest that the restoring effects of URB597 on the expression of mPFC miR-16 in males and miR-135a in females in stressed rats are not due to an additive effect.

#### 3.2.1. Effects on miRNAs and Their Targets in Females

ELS downregulated miR-135a in the mPFC and upregulated miR-135a in the LHb and DR with no significant effects on miR-16. Paroxetine downregulated miR-135a in the LHb and DR. Both URB597 and paroxetine upregulated miR-16 in the DR in stressed females.

mPFC-miR-135a downregulation was significantly correlated with hypo-activity and increased freezing in the OFT, suggesting that the downregulation is associated with the depressive-like phenotype. ELS also upregulated *htr1a, scl64a,* and *cnr2* and downregulated *cnr1* and *faah* mRNA in the mPFC. These effects were significantly correlated with the stressful phenotype, especially, upregulation of *cnr2* was significantly correlated with hypo-activity and elevated freezing in the OFT, and downregulation of *faah* was significantly correlated with decreased SP. Treatment with URB597 decreased *cnr2* expression compared to the ELS-Vehicle group and normalized *faah* expression. Moreover, we found significant negative correlations between miR-135a and cnr2 suggesting a possible association between them; this is the first study to examine the effects of URB597 on miR-135a and *cnr2*. Taken together, the findings may suggest that URB597 restored depressive-like behavior in females by increasing miR-135a and possibly CB2r in the mPFC. This corroborates with our recent study showing increased prelimbic *cnr2* expression in a rat model for schizophrenia that was restored by chronic treatment with URB597 [33]. Additionally, exposure to innate predator stress was shown to increase *cnr2* expression in the PFC [34].

Studies show alterations in FAAH gene expression associated with depressive-like phenotypes; in a genetic rat model of depression, the Flinders Sensitive Line, *faah* mRNA levels were decreased in the right PFC [35], however in males exposed to maternal deprivation, PFC-*faah* gene expression was increased [36]. In our study ELS decreased mPFC-*faah* expression in both sexes, however only in females, URB597 restored this effect. The decrease in mPFC-*faah* expression was significantly correlated with decreased SP, suggesting depressive-like behavior. These differences between males and females may be associated with estrogen, as miR-135a negatively targets the estrogen receptor alpha (ERα), one of the key regulators of estrogen signaling [37].

We also observed ELS-induced upregulation of miR-135a in the DR that was reduced with URB597 treatment. Other studies found alterations in miR-135a in the DR, showing upregulation following chronic restraint exposure [38] and downregulation following exposure to social defeat [26]. Importantly, we found that increased miR-135a expression was associated with decreased SP, hypo-activity, and increased freezing in the OFT, suggesting a depressive- and anxiogenic-like phenotype.

#### 3.2.2. Effects on miRNAs and Their Targets in Males

ELS downregulated miR-16 in the mPFC and miR-135a and miR-16 in the LHb, but not in the CA1 or DR in males. In stressed males, URB597 restored the expression of miR-16 in the mPFC; also, URB597 and paroxetine upregulated ELS-induced decrease in miR-16 in the LHb, and paroxetine upregulated miR-16 in the DR. Paroxetine was also found to downregulate miR-135a in males in all brain areas tested (mPFC, CA1, LHb and DR).

mPFC-miR-16 downregulation was significantly correlated with hypo-activity and increased freezing in the OFT. However, miR-16 downregulation was also correlated with elevated SP which was found to be intact in ELS males.

ELS downregulated mPFC mRNA expression of *htr1a, scl64a, cnr1, cnr2* and *faah*. Interestingly, these effects were not restored by treatment with URB597 and there was no correlation between the expression of miR-16 and these genes in the mPFC. This may suggest that the restoring effects of URB597 on mPFC-miR-16 in males are not mediated via its effects on serotonergic (5HT1A and SERT) mechanisms or its effects on *cnr1*, *cnr2* and *faah* mRNA. An interesting conclusion from these findings is that the restoring effects of URB597 in ELS males and females are mediated by different mechanisms.

ELS also downregulated miR-16 in the LHb, and treatment with URB597 and paroxetine elevated its expression. No correlation was observed between LHb-miR-16 expression and behavior.

One possible pathway for URB597 regulation of these miRNAs is through Wnt/ β-catenin. We have recently suggested a potentially novel mechanism for the stress-ameliorating effects of URB597 that involves the activation of CB1r and the Wnt/β-catenin pathway in the nucleus accumbens [39]. Specifically, we suggest that CB1r activation increases PI3K/AKT activity, which phosphorylates GSK-3β via Akt. Therefore, β-catenin is stabilized and translocated into the nucleus. Following this translocation, β-catenin regulates transcription and gene expression such as cyclin D1 which is involved in cell proliferation regulation [40]. As a result, β-catenin activates TCF/Lef transcription factors [41] and miRs [42] which promote a resilient response to stress. This mechanism could be a possible pathway mediating the anti-stress effects of URB597 on behavior and will be examined in future studies.

Women are twice as likely to develop depression following chronic stress than men and this sex difference indicates a potential role for gonadal hormones in the etiology of depressive disorders [43]. In support, studies suggest that mood disturbances in women are more frequent during periods of estrogen fluctuation or reduction [44]. Testosterone, on the other hand, has anxiolytic and antidepressant effects in women, men, and animals [45]. More research is needed to examine the mechanisms underlying the involvement of testosterone and estrogen in the antidepressant effects of cannabinoids. 

To summarize, this study demonstrates that exposure to ELS has deleterious protracted effects on behavior, mPFC-miRs and their targets in adulthood in both sexes. Although we found a similar behavioral phenotype for both sexes, sex-dependent alterations in mPFC-miRs and their targets were found. Specifically, ELS reduced miR-135a in females and miR-16 in males. FAAH inhibition by URB597 during post-adolescence restored these effects and the depressive-like behavioral phenotype. Moreover, this study suggests that in females, but not in males, the restoring effects of URB597 are mediated via CB2r signaling and normalization of *faah* expression. In males, these effects are mediated by a different mechanism.

## 4. Materials and Methods

### 4.1. Subjects

Sprague Dawley (SD) dams (Envigo, Israel) and pups were housed in polypropylene cages (59 × 28 × 20 cm^3^). Pups were weaned on P21 and separated randomly into equally sized (*n* = 5) groups according to sex at 22 ± 2 °C under 12 h light/dark cycles (lights turned on at 07:00). Food and water were available ad libitum. From each litter, no more than one male and one female were randomly assigned to each experimental condition (all pups were used in the different treatment groups). The experiments were approved by the University of Haifa Ethics and Animal Care Committee and appropriate measures were taken to minimize pain and discomfort (approval numbers: 745/20, 771/21).

### 4.2. Early Stress (ELS) Model

The early life stress was the LBN paradigm [30,31] with slight modifications [12,13,14]. The dam and her pups were housed in a cage with limited “Sunny Chips” bedding material (1.2 cm layer) from P7 to 14. The NoELS dam and her pups were housed in a cage with abundant (7–9 cm layer) bedding material.

### 4.3. Drugs

The FAAH inhibitor URB597 (0.4 mg/kg; i.p.; Cayman chemicals, Ann Arbor, MI, USA; [13]) was dissolved in dimethylsulfoxide (DMSO), saline (0.9% NaCl) and tween 80 (final DMSO concentration: <2%). Paroxetine hydrochloride (5 mg/kg; i.p.; Sigma-Aldrich, Jerusalem, Israel [21] was dissolved in saline. Controls were injected with vehicle (DMSO, saline-0.9% NaCl and tween 80; final DMSO concentration: <2%). The injections were performed on P45-60 (i.e., late-adolescence), based on previous studies [13,14,21,32].

### 4.4. Behavioral Tests

All rats were exposed to the same battery of behavioral tests in adulthood, 30 days after the last injection. Males and females were tested on separate days. The tests were carried out in the following order: activity and anxiety-like behavior in a novel open field arena, social tests (social preference and social recognition) and forced swim test (FST). The social tests were carried out in the open field arena (after 3 days of habituation to the arena). The most aversive test (FST) was last. Testing occurred under dim lighting (15–20 lx) and took place between 13:00 and 16:00 h.

#### 4.4.1. Activity and Anxiety-like Behavior in an Open Field (OFT)

The apparatus consists of a square black open field (50 × 50 × 50 cm^3^). The floor is divided by 1-cm-wide white lines into 25 squares measuring 10 × 10 cm^2^ each. The open field arena was thoroughly cleaned between each trial. The movements of the rat were recorded and analyzed for 30 min using a video tracking system (Ethovision ×T 14.0, Noldus Information Technology) to measure locomotion (measured as distance moved in cm). Freezing time (s) and time spent in the center (s) in the first 5 min were used as a measure of anxiety-like behavior [33,46].

#### 4.4.2. Social Preference (SP) and Social Recognition (SR)

This task aims to assess sociability and short-term social memory. The “partner” rat was confined to a separate section of the open field (50 × 50 × 50 cm^3^) by a transparent perforated Plexiglas panel (corrals; [14]). For the preference phase, the experiment rat was given 5 min exploration with a novel juvenile and a novel object. For the recognition phase, after 30 min in a holding cage, the rat was given 5 min exploration with the familiar juvenile and a novel juvenile, both confined to the corrals. The trials were videotaped (Dericam, Indoor Pan/tilt IP camera M801W, USA). An exploration index was calculated; for social preference: time exploring the novel juvenile/ total exploration time (object + juvenile rat). For social recognition: time exploring the novel juvenile/total exploration time (familiar + novel juvenile).

#### 4.4.3. Forced Swim Test (FST)

The test is based on the assumption that when an animal is placed in a container filled with water, it will first make efforts to escape but eventually will exhibit immobility that may reflect despair [47]. A cylindrical water container (62 cm diameter, 40 cm height, filled with a water at temperature of 23 °C) was used. The water level was such that the rat could not touch the bottom with its hind paws. Two swimming sessions were conducted: a 15 min pretest (the learning phase) followed by a 5 min test the next day. Video films (Logitech C922 1920 × 1080, Lausanne Switzerland) of the second day of each FST session were analyzed for immobility time in s [48].

### 4.5. Real-Time (RT) PCR

Rats were sacrificed and brain tissues of the mPFC, CA1, LHB and DR were harvested for molecular analysis (see Appendix A). RNA extraction, cDNA preparation and qRT-PCR were performed as previously described [49] to detect the expression of miRNAs (miR-135a and miR-16) and mRNA of serotonergic and ECB markers. We examined the ECB markers cannabinoid receptors 1 and 2 (*cnr1* and *cnr2*, respectively), and FAAH (*faah*) as well as the serotonergic markers 5HT1A (*htr1a*) and SERT (*slc6a4*). For mRNA, 1000 ng of total RNA was converted into cDNA using qScript cDNA Synthesis Kit (Quanta Biosciences, Gaithersburg, USA). For miRNA, 500 ng of total RNA was reverse transcribed cDNA using qScript microRNA cDNA Synthesis Kit (Quanta Biosciences, Gaithersburg, MD, USA). This was followed by Real-Time SYBR Green qRT-PCR amplification using specific primers (Quanta Biosciences, Gaithersburg, USA) according to the manufacturer’s instructions. RT reactions were carried out by a Step One real-time PCR system (Applied Biosystems). Fold-change values were calculated using the ddCt method relative to the housekeeping gene hypoxanthine phosphoribosyl transferase (HPRT; mRNA) or RNU6 (miRNA). Primers for both miRNAs (miR-135a-5p and miR-16-5p) and mRNAs (see Table 5) were designed and synthesized by Agentek (Tel Aviv, Israel). Primer suitability was determined using standard curve analysis, melting curve analysis and linearity and threshold [50].

### 4.6. Statistical Analysis

The results are expressed as means ± SEM. For statistical analysis, one-way ANOVA, two-way ANOVA, three-way ANOVA, one-sample *t*-test, independent-samples *t*-test and Pearson bivariate correlation test were used as indicated. All post hoc comparisons were made using Tukey’s range test. Significance was set at *p* ≤ 0.05. Based on previous studies which demonstrate sex differences in stress response, behavior, neuroendocrine system, gene methylation and MDD pathophysiology [12,13,14,24], we performed statistical analysis for each sex separately. Data were analyzed using SPSS 27 (IBM, Chicago, IL, USA). Normality assumption was examined using the Kolmogorov–Smirnov and Shapiro–Wilk tests.

## 5. Conclusions

As far as we know our findings show for the first time that FAAH inhibition can prevent an ELS-induced decrease in mPFC miRs and the associated stressful phenotype in a sex-dependent manner. Although males and females demonstrated stressful behaviors following ELS, the stressful phenotype is different as well as the alterations in miRs and their response to treatment with URB597. Exploring the differential effects of stress and drugs in males and females is crucial for the development of personalized, sex-specific approaches for treatment. Moreover, our findings advance our knowledge on dysfunctional pathways in depression in cortical areas and suggest a mechanism for the beneficial effects of enhancing endocannabinoid signaling.

## Figures and Tables

**Figure 1 ijms-23-16101-f001:**
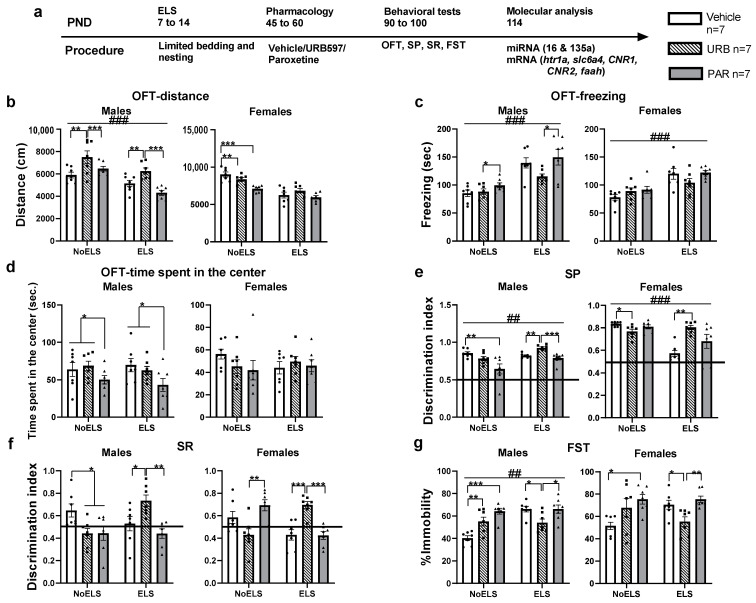
The effects of ELS and chronic late-adolescence treatment with URB or PAR on depressive-like behavior in adult males and females. (**a**) Male and female rats were exposed to early life stress (ELS; P7-14). Vehicle, URB597 (URB) or paroxetine (PAR) were injected during late adolescence (P45-60). Behavioral testing started on P90, and rats were sacrificed on P114 for molecular analysis (*n* = 7 for all groups). (**b**) In the OFT, ELS decreased the distance travelled in males, and URB treatment increased it compared to groups treated with vehicle and URB (left). PAR treatment decreased the distance travelled in NoELS females compared to females treated with vehicle and URB (right). (**c**) ELS increased freezing time in males; PAR treatment increased freezing time compared to URB treatment (left). In females, ELS increased freezing time (right). (**d**) PAR treatment decreased time spent in the center compared to vehicle and URB treatments in males (left). No significant differences were observed between the groups in females (right). (**e**) In the SP test, in males (left), URB treatment increased SP in the ELS groups compared to vehicle and PAR treatment; PAR treatment decreased SP compared to vehicle treatment in the NoELS groups (left). In ELS females (right), URB treatment increased SP compared to vehicle treatment; URB treatment decreased SP compared to vehicle treatment in NoELS females. (**f**) In the SR test, in ELS males (left), URB treatment increased the SR discrimination index compared to vehicle and PAR treatments; in NoELS males, URB and PAR treatment decreased SR compared to vehicle treatment (left). In ELS females (right), URB treatment increased the SR discrimination index compared to vehicle and PAR treatment; PAR treatment increased the SR discrimination index compared to the URB treatment in NoELS females. (**g**) In the FST, in ELS males (left), URB treatment decreased immobility compared to vehicle and PAR treatments; in NoELS males, vehicle treatment decreased immobility compared to URB and PAR treatments. In ELS females (right), URB treatment decreased immobility compared to the vehicle and PAR treatments; in NoELS females, PAR treatment increased immobility compared to vehicle treatment. ELS: early life stress; FST: forced swim test; miRNA: microRNA; mRNA: messenger RNA; OFT: open-field test; PND: post-natal day; SP: social preference; SR: social recognition. *, *p <* 0.05; **, *p <* 0.01; ***, *p <* 0.001 indicate statistically significant effects followed by post hoc comparisons; ##, *p <* 0.01; ###, *p <* 0.001 indicate statistical significance in main effects.

**Figure 2 ijms-23-16101-f002:**
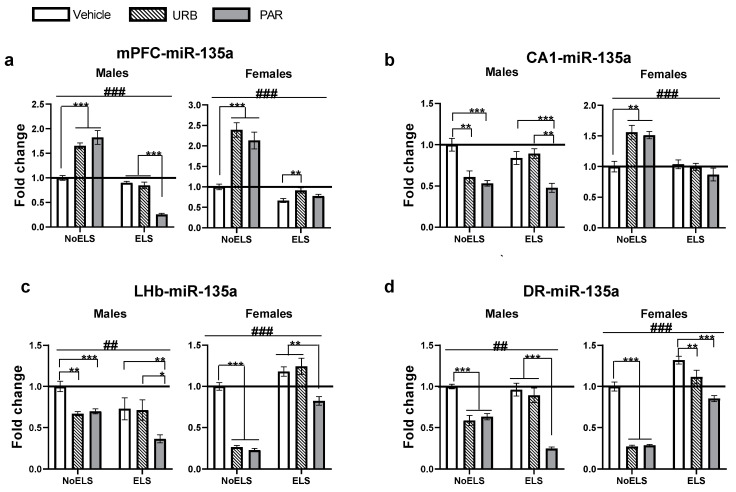
The effects of ELS and chronic late-adolescence treatment with URB or PAR on the expression of miR-135a-5p in adult males and females. (**a**) In the mPFC, PAR downregulated the expression of miR-135a in ELS males compared to ELS-Vehicle and ELS-URB (left). In NoELS males (left) and females (right), URB and PAR upregulated the expression compared to NoELS-Vehicle. In females (right), URB normalized the expression of miR-135a compared to ELS-Vehicle (*n* = 5–9). (**b**) In the CA1, in ELS males (left), PAR downregulated expression compared to ELS-Vehicle or ELS-URB; in NoELS males (left) both URB and PAR downregulated expression. In NoELS females (right), both URB and PAR upregulated expression (*n* = 5–9). (**c**) In the LHb, in ELS males (left) and females (right), PAR downregulated expression compared to ELS-Vehicle males and females or URB. In NoELS males and females, URB and PAR downregulated its expression (*n* = 5–9). (**d**) In the DR, in ELS males (left), PAR downregulated expression compared to ELS-Vehicle and ELS-URB. In NoELS males (left) and females (right), URB and PAR downregulated expression compared to NoELS-Vehicle males and females. In ELS females (right), URB and PAR downregulated its expression compared to ELS-Vehicle (*n* = 5–8). DR: dorsal raphe; ELS: early life stress; LHb: lateral habenula; mPFC: medial prefrontal cortex; PAR: paroxetine; URB: URB597. *, *p <* 0.05; **, *p <* 0.01; ***, *p <* 0.001 indicate statistically significant effects followed by post hoc comparisons. ##, *p <* 0.01; ###, *p <* 0.001 indicate statistical significance in main effects.

**Figure 3 ijms-23-16101-f003:**
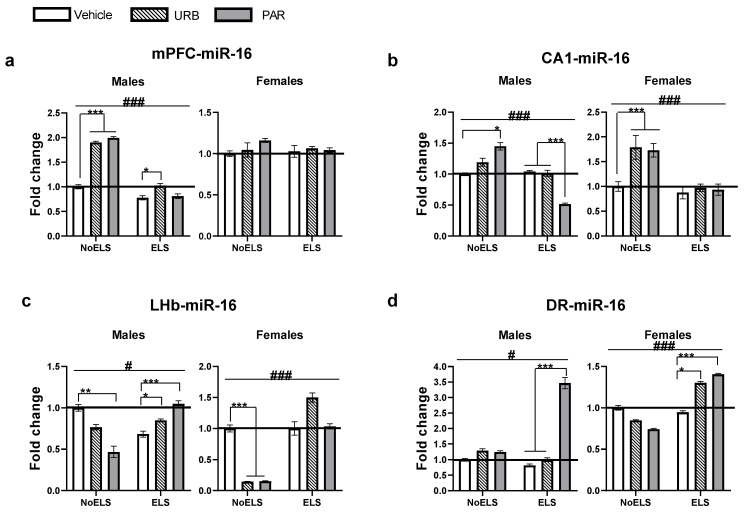
The effects of ELS and chronic late-adolescence treatment with URB or PAR on the expression of miR-16-5p in adult males and females. (**a**) In the mPFC, in ELS males, URB normalized the expression of miR-16 compared to ELS-Vehicle; in NoELS males, URB and PAR upregulated expression compared to NoELS-Vehicle. No significant differences were observed in females (*n* = 5–10). (**b**) In the CA1, in ELS males, PAR downregulated expression compared ELS-Vehicle or ELS-URB; in NoELS males PAR upregulated expression compared to NoELS-Vehicle; in NoELS females, URB and PAR upregulated expression compared to NoELS-Vehicle (*n* = 5–8). (**c**) In the LHb, in ELS males, URB and PAR upregulated expression compared to ELS-Vehicle; in NoELS males, PAR downregulated expression compared to NoELS-Vehicle. In NoELS females, URB and PAR downregulated expression compared to NoELS-Vehicle (*n* = 5–8). (**d**) In the DR, in ELS males, PAR upregulated expression compared to ELS-Vehicle or ELS-URB. In ELS females, URB and PAR upregulated expression compared to ELS-Vehicle (*n* = 5–8). DR: dorsal raphe; ELS: early life stress; LHb: lateral habenula; mPFC: medial prefrontal cortex; PAR: paroxetine; URB: URB597. *, *p <* 0.05; **, *p <* 0.01; ***, *p <* 0.001 indicate statistically significant effects followed by post hoc comparisons. #, *p <* 0.05; ###, *p <* 0.001 indicate statistical significance in main effects.

**Figure 4 ijms-23-16101-f004:**
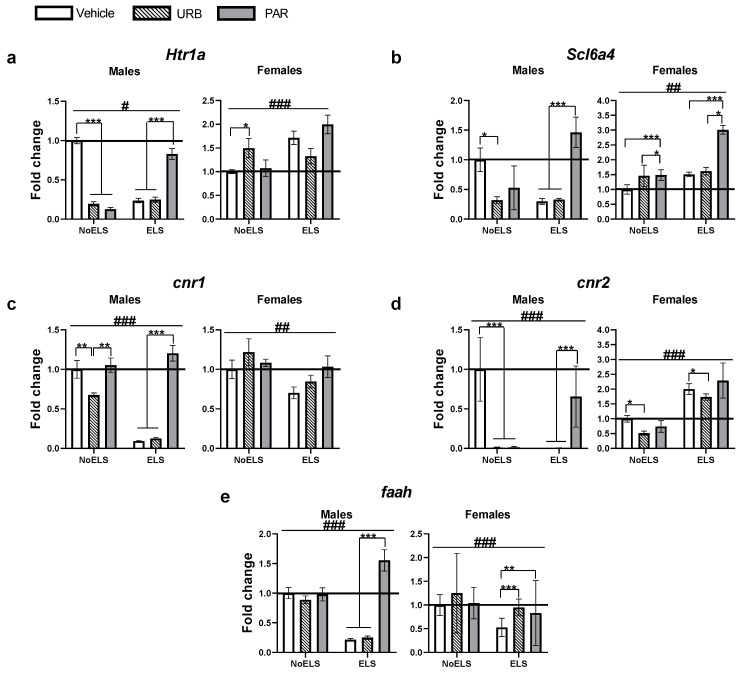
The effects of ELS and chronic late-adolescence treatment with URB or PAR on mRNA expression of serotonergic and endocannabinoid genes in the mPFC in adult males and females. (**a**) In ELS males, URB or vehicle downregulated the expression of *htr1a* compared to ELS-PAR; in NoELS males, URB and PAR downregulated its expression compared to NoELS-Vehicle. In females, URB upregulated *htr1a* expression compared to NoELS-Vehicle (*n* = 5–8). (**b**) In ELS males, vehicle and URB downregulated the expression of *slc6a4* compared to ELS-PAR; in NoELS males, URB downregulated its expression compared to NoELS-Vehicle. In ELS and NoELS females, PAR upregulated the expression of *slc6a4* compared to ELS- and NoELS-Vehicle or URB (*n* = 5–6). (**c**) In ELS males, URB or vehicle downregulated the expression of *cnr1* compared to ELS-PAR; in NoELS males, URB downregulated its expression compared to NoELS-Vehicle or NoELS-PAR. In females, ELS downregulated *cnr1* expression compared to the NoELS groups (*n* = 5–10). (**d**) In ELS males, URB or vehicle downregulated *cnr2* expression compared to ELS-PAR; in NoELS males, URB or PAR downregulated its expression compared to NoELS-Vehicle. In females, ELS upregulated the expression of *cnr2* and URB decreased *cnr2* expression compared to ELS- and NoELS-Vehicle or PAR (*n* = 5–8). (**e**) In ELS males, URB or vehicle downregulated *faah* expression compared to ELS-PAR. In females, URB or PAR upregulated the expression of *faah* compared to ELS-Vehicle (*n* = 5–7). ELS: early life stress; PAR: paroxetine; URB: URB597. *, *p <* 0.05; **, *p <* 0.01; ***, *p <* 0.001 indicate statistically significant effects followed by posthoc comparisons; #, *p <* 0.05; ##, *p <* 0.01; ###, *p <* 0.001 indicate statistical significance in main effects.

**Table 1 ijms-23-16101-t001:** Correlations between the expression of miR-135a-5p and behavior in adult males and females.

	SP	SR	FST- Immobility	OF- Distance	OF- Freezing	OF- Center
mPFC	Males:	Males:	Males:	Males:	Males:	Males:
	r = −0.34;	r = 0.46;	r = −0.19;	r = 0.62;	r = −0.57;	r = 0.16;
	*p* = 0.082	*p* = 0.831	*p* = 0.330	*p* = 0.001	*p* = 0.002	*p* = 0.423
	Females:	Females:	Females:	Females:	Females:	Females:
	r = 0.34;	r = 0.24;	r = 0.18;	r = 0.35;	r = −0.37;	r = 0.38;
	*p* = 0.057	*p* = 0.184	*p* = 0.312	*p* = 0.043	*p* = 0.030	*p* = 0.834
CA1	Males:	Males:	Males:	Males:	Males:	Males:
	r = 0.30;	r = 0.51;	r = −0.25;	r = 0.12;	r = −0.20;	r = 0.40;
	*p* = 0.110	*p* = 0.003	*p* = 0.205	*p* = 0.512	*p* = 0.274	*p* = 0.022
	Females:	Females:	Females:	Females:	Females:	Females:
	r = 0.74;	r = 0.10;	r = 0.43;	r = 0.21;	r = −0.25;	r = 0.20;
	*p* = 0.693	*p* = 0.600	*p* = 0.826	*p* = 0.269	*p* = 0.200	*p* = 0.291
LHb	Males:	Males:	Males:	Males:	Males:	Males:
	r = 0.21;	r = 0.41;	r = −0.27;	r = 0.36;	r = −0.64;	r = 0.36;
	*p* = 0.283	*p* = 0.024	*p* = 0.219	*p* = 0.052	*p* < 0.001	*p* = 0.049
	Females:	Females:	Females:	Females:	Females:	Females:
	r = −0.31;	r = −0.17;	r = −0.29;	r = −0.28;	r = 0.32;	r = 0.36;
	*p* = 0.145	*p* = 0.376	*p* = 0.131	*p* = 0.117	*p* = 0.074	*p* = 0.045
DR	Males:	Males:	Males:	Males:	Males:	Males:
	r = 0.17;	r = 0.41;	r = −0.36;	r = 0.48;	r = 0.09;	r = 0.51;
	*p* = 0.364	*p* = 0.025	*p* = 0.065	*p* = 0.006	*p* = 0.635	*p* = 0.004
	Females:	Females:	Females:	Females:	Females:	Females:
	r = −0.50;	r = −0.09;	r = 0.14;	r = −0.48;	r = 0.45;	r = 0.07;
	*p* = 0.010	*p* = 0.675	*p* = 0.506	*p* = 0.016	*p* = 0.025	*p* = 0.731

DR: dorsal raphe; FST: forced swim test; LHb: lateral habenula; mPFC: medial prefrontal cortex; OF: open field test; SP: social preference; SR: social recognition.

**Table 2 ijms-23-16101-t002:** Correlations between the expression of miR-16-5p and behavior in adult males and females.

	SP	SR	FST- Immobility	OF- Distance	OF- Freezing	OF- Center
mPFC	Males:	Males:	Males:	Males:	Males:	Males:
	r = −0.45;	r = −0.01;	r = −0.002;	r = 0.67;	r = −0.57;	r = −0.05;
	*p* = 0.017	*p* = 0.936	*p* = 0.990	*p* < 0.001	*p* = 0.001	*p* = 0.764
	Females:	Females:	Females:	Females:	Females:	Females:
	r = 0.31;	r = −0.01;	r = −0.001;	r = −0.06;	r = −0.01;	r = 0.61;
	*p* = 0.131	*p* = 0.947	*p* = 0.994	*p* = 0.755	*p* = 0.941	*p* = 0.768
CA1	Males:	Males:	Males:	Males:	Males:	Males:
	r = −0.38;	r = 0.32;	r = −0.007;	r = 0.55;	r = −0.40;	r = 0.07;
	*p* = 0.043	*p* = 0.082	*p* = 0.969	*p* = 0.001	*p* = 0.018	*p* = 0.660
	Females:	Females:	Females:	Females:	Females:	Females:
	r = 0.31;	r = 0.08;	r = 0.20;	r = 0.34;	r = −0.47;	r = 0.04;
	*p* = 0.104	*p* = 0.667	*p* = 0.335	*p* = 0.082	*p* = 0.015	*p* = 0.826
LHb	Males:	Males:	Males:	Males:	Males:	Males:
	r = 0.18;	r = 0.03;	r = −0.25;	r = −0.37;	r = 0.26;	r = −0.05;
	*p* = 0.371	*p* = 0.879	*p* = 0.184	*p* = 0.061	*p* = 0.190	*p* = 0.793
	Females:	Females:	Females:	Females:	Females:	Females:
	r = −0.14;	r = 0.03;	r = −0.17;	r = −0.42;	r = 0.46;	r = 0.27;
	*p* = 0.488	*p* = 0.852	*p* = 0.391	*p* = 0.022	*p* = 0.011	*p* = 0.152
DR	Males:	Males:	Males:	Males:	Males:	Males:
	r = −0.19;	r = −0.46;	r = 0.015;	r = −0.40;	r = −0.13;	r = −0.28;
	*p* = 0.283	*p* = 0.007	*p* = 0.938	*p* = 0.019	*p* = 0.439	*p* = 0.109
	Females:	Females:	Females:	Females:	Females:	Females:
	r = −0.07;	r = −0.09;	r = −0.20;	r = −0.55;	r = 0.53;	r = 0.13;
	*p* = 0.735	*p* = 0.641	*p* = 0.317	*p* = 0.003	*p* = 0.005	*p* = 0.504

DR: dorsal raphe; FST: forced swim test; LHb: lateral habenula; mPFC: medial prefrontal cortex; OF: open field test; SP: social preference; SR: social recognition.

**Table 3 ijms-23-16101-t003:** Correlations between the expression of serotonergic and ECB mRNAs in the mPFC and behavior in adult males and females.

	SP	SR	FST- Immobility	OF- Distance	OF- Freezing	OF- Center
*htr1a*	Males:	Males:	Males:	Males:	Males:	Males:
	r = 0.364;	r = 0.50;	r = −0.65;	r = −0.62;	r = 0.11;	r = −0.07;
	*p* = 0.088	*p* = 0.011	*p* = 0.001	*p* = 0.002	*p* = 0.962	*p* = 0.728
	Females:	Females:	Females:	Females:	Females:	Females:
	r = −0.46;	r = −0.41;	r = 0.28;	r = −0.30;	r = 0.26;	r = −0.28;
	*p* = 0.018	*p* = 0.077	*p* = 0.237	*p* = 0.188	*p* = 0.255	*p* = 0.227
*scl6a4*	Males:	Males:	Males:	Males:	Males:	Males:
	r = −0.21;	r = −0.36;	r = −0.40;	r = −0.28;	r = −0.008;	r = 0.06;
	*p* = 0.383	*p* = 0.109	*p* = 0.062	*p* = 221	*p* = 0.973	*p* = 0.790
	Females:	Females:	Females:	Females:	Females:	Females:
	r = −0.46;	r = −0.04;	r = 0.45;	r = −0.41;	r = 0.46;	r = 0.28;
	*p* = 0.041	*p* = 0.858	*p* = 0.052	*p* = 0.069	*p* = 0.040	*p* = 0.220
*cnr1*	Males:	Males:	Males:	Males:	Males:	Males:
	r = −0.47;	r = −0.56;	r = −0.15;	r = 0.07;	r = −0.24;	r = −0.27;
	*p* = 0.020	*p* = 0.003	*p* = 0.432	*p* = 0.697	*p* = 0.225	*p* = 0.159
	Females:	Females:	Females:	Females:	Females:	Females:
	r = −0.005;	r = 0.14;	r = 0.36;	r = 0.30;	r = −0.31;	r = −0.22;
	*p* = 0.979	*p* = 0.464	*p* = 0.064	*p* = 0.110	*p* = 0.104	*p* = 0.260
*cnr2*	Males:	Males:	Males:	Males:	Males:	Males:
	r = −0.03;	r = −0.15;	r = −0.35;	r = −0.30;	r = −0.21;	r = −0.13;
	*p* = 0.875	*p* = 0.464	*p* = 0.061	*p* = 0.127	*p* = 0.298	*p* = 0.526
	Females:	Females:	Females:	Females:	Females:	Females:
	r = −0.31;	r = −0.27;	r = 0.13;	r = −0.53;	r = 0.44;	r = −0.12;
	*p* = 0.110	*p* = 0.202	*p* = 0.585	*p* = 0.008	*p* = 0.036	*p* = 0.563
*faah*	Males:	Males:	Males:	Males:	Males:	Males:
	r = −0.49;	r = −0.52;	r = −0.12;	r = −0.11;	r = −0.07;	r = −0.34;
	*p* = 0.019	*p* = 0.008	*p* = 545	*p* = 0.587	*p* = 0.714	*p* = 0.082
	Females:	Females:	Females:	Females:	Females:	Females:
	r = 0.46;	r = 0.29;	r = 0.09;	r = 0.21;	r = 0.30;	r = −0.39;
	*p* = 0.025	*p* = 0.158	*p* = 0.678	*p* = 0.273	*p* = 0.159	*p* = 0.062

FST: forced swim test; OF: open field test; SP: social preference; SR: social recognition. *Htr1a* for 5HT1Ar, *slc6a4* for SERT, *cnr1* for CB1r, *cnr2* for CB2r and *faah* for FAAH.

**Table 4 ijms-23-16101-t004:** Correlations between the expression of miRNA (miR-135a & miR-16) and possible mRNA targets (serotonergic and ECB) in the mPFC in adult males and females.

	*h* *tr1a*	*s* *cl6a4*	*cnr* *1*	*cnr* *2*	*f* *aah*	miR-135a	miR-16
miR-135a	Males:	Males:	Males:	Males:	Males:		Males:
	r = −0.55;	r = −0.004;	r = −0.07;	r = −0.51;	r = −0.22;		r = 0.71;
	*p* = 0.005	*p* = 0.985	*p* = 0.716	*p* = 0.012	*p* = 0.294		*p* < 0.001
	Females:	Females:	Females:	Females:	Females:		Females:
	r = −0.21;	r = 0.25;	r = 0.28;	r = −0.76;	r = 0.54;		r = 0.29;
	*p* = 0.330	*p* = 0.239	*p* = 0.108	*p* < 0.001	*p* = 0.004		*p* = 0.078
miR-16	Males:	Males:	Males:	Males:	Males:	Males:	
	r = −0.32;	r = −0.22;	r = 0.34;	r = −0.21;	r = 0.20;	r = 0.71;	
	*p* = 0.142	*p* = 0.342	*p* = 0.073	*p* = 0.300	*p* = 0.308	*p* < 0.001	
	Females:	Females:	Females:	Females:	Females:	Females:	
	r = −0.39;	r = −0.49;	r = 0.005;	r = −0.23;	r = −0.21;	r = 0.29;	
	*p* = 0.060	*p* = 0.018	*p* = 0.980	*p* = 0.239	*p* = 0.273	*p* = 0.078	

*Htr1a* for 5HT1Ar, *slc6a4* for SERT, *cnr1* for CB1r, *cnr2* for CB2r and *faah* for FAAH.

**Table 5 ijms-23-16101-t005:** Primers for mRNAs used for real-time PCR.

Name	Description	Gene Bank ID (NM)	Protein Name	Primer Sequence	Efficacy (%)
*h* *prt*	Housekeeping gene; used as a reference gene	NM_012583.2	HPRT	F: 5′CGCCAGCTTCCTCCTCAG3′ R: 5′ATAACCTGGTTCATCATCACTAATCAC3′	99.83
*h* *tr1a*	Serotonergic auto-receptor	NM_012585.1	5HT1A	F: 5′CCACGGCTACACCATCTACTC3′ R: 5′AAGCGTGCGGCTCTGAAG3′	96.67
*s* *lc6a4*	The serotonergic transporter	NM_013034.4	SERT	F: 5′CAGCCCTCTGTTTCTCCTGTTC3′ R: 5′CCTATGCAGTAGCCCAAGACGA3′	79.46
*cnr* *1*	Cannabinoid receptor 1	NM_012784.5	CB1	F: 5′CACCCATGGCTGAGGGTTC3′ R: 5′CTGCAAGGCCATCTAGGATCGA3′	99.27
*cnr* *2*	Cannabinoid receptor 2	NM_020543.4	CB2	F: 5′GCCTGCAACTTCGTCATCTTC3′ R: 5′TGCCGATCTTCAACAGGAA3′	118.04
*f* *aah*	The enzyme responsible for AEA degradation	NM_024132.3	FAAH	F: 5′TGCCCTTCAGAGAGGAGGT3′ R: 5′CTGGGCATGGTATAGTTGTCAGT3′	107.92

F: forward primer; R: reverse primer.

## Data Availability

Not applicable.

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
