# Peer review of "FAAH Inhibition Restores Early Life Stress-Induced Alterations in PFC microRNAs Associated with Depressive-Like Behavior in Male and Female Rats"

_ijms, 2022, doi:10.3390/ijms232416101_

Round 1

Reviewer 1 Report

The authors have attempted to evaluate the effects of a novel FAAH inhibitor on the neurobiological effects of early life stress (ELS) in laboratory rats. The following are some of the comments & observations:

Abstract:

The abstract needs to mention that it was a study performed on "lab rats". The research subject population is missing in the abstract provided. 

Methods:

Why is this section written at the end? Shouldn't it appear before the results?

Results:

The results section is quite exhaustive though restricted to answering the primary and secondary objectives of the study. However, to make it more comprehensible, the sub-sections within the results need to be highlighted (Eg: by boldening the text). 

Discussion:

No comments. 

Author Response

    1. Abstract: the abstract needs to mention that it was a study performed on "lab rats". The research subject population is missing in the abstract provided. 

    Added:

    Abstract: We compared the effects of treatment with the fatty acid amide hydrolase (FAAH) inhibitor URB597, and the selective serotonin reuptake inhibitor paroxetine, on ELS-induced depressive-like behavior and the expression of microRNAs (miRs) associated with depression in the medial prefrontal cortex (mPFC), hippocampal CA1 area, lateral habenula and dorsal raphe in rats.

    1. Methods: why is this section written at the end? Shouldn't it appear before the results?

    Corrected.

    1. Results: the results section is quite exhaustive though restricted to answering the primary and secondary objectives of the study. However, to make it more comprehensible, the sub-sections within the results need to be highlighted (Eg: by boldening the text).

    Corrected.

Reviewer 2 Report

The Authors of the manuscript described that URB597 reversed ELS-induced mPFC downregulation. Interestingly they associated effects with the expression of specific miRNAs. The manuscript is not easy to follow because the explanation of the results lists a number of data from which a summation is complex. The Authors have tried to make it easier to read by concluding each paragraph of the results with a general summary. A problem with the work is that it is mainly based on correlations.

For example, the Authors say "...we found significant negative correlations between miR-135a and cnr2; this may suggest that cnr2 is 638 a possible target of miR-135a."

Are binding sites for miR-135a in the cnr2 transcript? Do other miRNAs discussed in the manuscript present binding sites in the genes the Authors identified as altered by the treatments?

It would be more interesting to evaluate the possible functions of the miRNAs studied according to their potential targets and describe them. How do the Authors explain the regulation of these miRNAs with their treatments?

In the methods section, Authors say that they tested primers used in real-time PCR experiments, but results regarding their efficiency are not shown. Please include them.

Author Response

  1. A problem with the work is that it is mainly based on correlations. For example, the Authors say "...we found significant negative correlations between miR-135a and cnr2; this may suggest that cnr2 is 638 a possible target of miR-135a." Are binding sites for miR-135a in the cnr2 transcript? Do other miRNAs discussed in the manuscript present binding sites in the genes the Authors identified as altered by the treatments?

We deleted this sentence and added that these correlations may suggest a possible association between the miRs:

Results, line 710: These correlations may suggest a possible association between the miRs.

Discussion, line 761: Moreover, we found significant negative correlations between miR-135a and cnr2 suggesting a possible association between them; this is the first study to examine the effects of URB597 on miR-135a and cnr2.                        

  1. It would be more interesting to evaluate the possible functions of the miRNAs studied according to their potential targets and describe them. How do the Authors explain the regulation of these miRNAs with their treatments?

We added an explanation for the regulation of these miRNAs with the treatments (Discussion, line 805): 

One possible pathway for URB597 regulation of these miRNAs is through Wnt/ β-catenin.   We have recently suggested a potentially novel mechanism for the stress-ameliorating effects of URB597 that involves the activation of CB1r and the Wnt/β-catenin pathway in the nucleus accumbens [47]. Specifically, we suggest that CB1r activation increases PI3K/AKT activity, which phosphorylates GSK-3β via Akt. Therefore, β-catenin is stabilized and translocated into the nucleus. Following this translocation, β-catenin regulates transcription and gene expression such as cyclin D1 which is involved in cell proliferation regulation [48]. As a result, β-catenin activates TCF/Lef transcription factors [49] and miRs [50] promoting a resilient response to stress. This mechanism could be a possible pathway mediating the anti-stress effects of URB597 on behavior and will be examined in future studies.

  1. In the methods section, Authors say that they tested primers used in real-time PCR experiments, but results regarding their efficiency are not shown. Please include them.

We added a column with the efficiencies for each primer to table 1 in the methods section.

Reviewer 3 Report

The manuscript is very well-written, and interesting to read. This study shows that FAAH inhibition can prevent an ELS-induced decrease in mPFC and the associated stressful phenotype in a sex-dependent manner, and the differential effects of ELS and URB597 on males and females highlight the importance of developing sex-specific approaches for stress-related psychiatric disorders. The Figures, Figure legends, and Tables present a large amount of information efficiently and make the manuscript more comprehensible even though the format is a little bit different from the other papers. The methods and statistical analysis are well-conducted. I think the paper is good to publish, however, I hope all the authors could revise and edit the paper, there are some errors in the manuscript in its current format, such as the reader don't know what is CA1 and cnr2 in the abstract, also, something like 'FAAH' and 'faah' in the abstract.

Author Response

I think the paper is good to publish, however, I hope all the authors could revise and edit the paper, there are some errors in the manuscript in its current format, such as the reader don't know what is CA1 and cnr2 in the abstract, also, something like 'FAAH' and 'faah' in the abstract.

Corrected.

Abstract: …. associated with depression in the medial pre-frontal cortex (mPFC), hippocampal CA1 area, lateral habenula and dorsal raphe in rats. We also examined the mRNA expression of serotonergic (htr1a and slc6a4) and endocannabinoid (cnr1, cnr2 and faah) targets in the mPFC following ELS and pharmacological treatment.

Reviewer 4 Report

The original research manuscript deals with an interesting topic about the effects of paroxetine (a standard antidepressant drug) and a fatty acid amide hydrolase inhibitor on PFC microRNAs associated with depressive-like behaviours, especially induced by an early life stress. The authors explored the early life stressed male and female rats. Since early life stress induced behaviour impairments cause serious psychiatric diseases, the topic of this manuscript is important and timely.

The manuscript is well written. The experimental design is appropriate, and the used methods are adequate. The abstract as well as Introduction offer sufficient background. Furthermore, the results are supported by the data and properly discussed and presented in illustrative graphs or tables.

Nevertheless, I recommend a brief discussion about sexual hormones (e.g. oestrogen and testosterone) as well as reactive oxygen species on the stress signal cascade associated with depression pathogenesis.

Author Response

I recommend a brief discussion about sexual hormones (e.g. oestrogen and testosterone) as well as reactive oxygen species on the stress signal cascade associated with depression pathogenesis.

We added a paragraph in the discussion section regarding sexual hormones (Discussion, line 817): 

Women are twice as likely to develop depression following chronic stress than men and this sex difference indicates a potential role for gonadal hormones in the etiology of depressive disorders [51]. In support, studies suggest that mood disturbances in women are more frequent during periods of estrogen fluctuation or reduction [52]. Testosterone, on the other hand, has anxiolytic and antidepressant effects in women, men, and animals [53]. More research is needed to examine the mechanisms underlying the involvement of testosterone and estrogen in the antidepressant effects of cannabinoids.

There is also a paragraph discussing miR-135 and the estrogen receptor alpha (Discussion, line 773):

The decrease in mPFC- faah expression was significantly correlated with decreased SP, suggesting depressive-like behavior. These differences between males and females may be associated with estrogen, as miR-135a negatively targets the estrogen receptor alpha (ERα), one of the key regulators of estrogen signaling [45].

As for the reactive oxygen species, we believe that this topic is beyond the scope of our paper.

Round 2

Reviewer 2 Report

The authors responded satisfactorily to my misgivings about the initial version of the paper. I believe the work can be published as is.